# Ameliorative effects of elderberry (*Sambucus nigra L.*) extract and extract-derived monosaccharide-amino acid on $H_2O_2$-induced decrease in testosterone-deficiency syndrome in a TM3 Leydig cell

Sujung Lee[1,2], Jiyeon Kim[3], Hyunseok Kong[4,5]*, Yong-Suk Kim[2]*

1 Gochang Food & Industry Institute, Gochang, Korea, 2 Department of Food Science & Technology Jeonbuk National University, Jeonju, Korea, 3 KOSA BIO Inc., Namyangju-si, Korea, 4 College of Animal Biotechnology and Resource, Sahmyook University, Seoul, Korea, 5 PADAM Natural Material Research Institute, Sahmyook University, Seoul, Korea

* kimys08@jbnu.ac.kr (Y-SK); hskong0813@gmail.com (HK)

**Data Availability Statement:** All relevant data are within the manuscript and its Supporting Information files.

## Abstract

With aging, men develop testosterone-deficiency syndrome (TDS). The development is closely associated with age-related mitochondrial dysfunction of Leydig cell and oxidative stress-induced reactive oxygen species (ROS). Testosterone-replacement therapy (TRT) is used to improve the symptoms of TDS. However, due to its various side effects, research on functional ingredients derived from natural products that do not have side effects is urgently needed. In this study, using the mitochondrial dysfunction TM3 (mouse Leydig) cells, in which testosterone biosynthesis is reduced by $H_2O_2$, we evaluated the effects of elderberry extract and monosaccharide-amino acid (fructose–leucine; FL) on mRNA and protein levels related to steroidogenesis-related enzymes steroidogenic acute regulatory protein (StAR), cytochrome P450 11A1(CYP11A1, cytochrome P450 17A1(CYP17A1), cytochrome P450 19A1(CYP19A1, aromatase), 3β-hydroxysteroid dehydrogenase (3β-HSD), and 17β-hydroxysteroid dehydrogenase(17β-HSD). We analyzed elderberry extract and extract-derived FL for changes in ROS scavenging activity and testosterone secretion. Elderberry extract and FL significantly reduced $H_2O_2$-induced intracellular ROS levels, improved testosterone secretion, and increased the mRNA and protein expression levels of steroidogenesis-related enzymes (StAR, 3b-HSD, 17b-HSD, CYP11A1, CYp17A1). However, the conversion of testosterone to estradiol was inhibited by elderberry extract and extract-derived FL, which reduced the mRNA and protein expression of CYP19A1. In conclusion, elderberry extract and FL are predicted to have value as novel functional ingredients that may contribute to the prevention of TDS by ameliorating reduced steroidogenesis.

**Funding:** This research was supported by a grant from the Technology Development Program (S2778672) funded by the Korea Technology & Infor-mation Promotion Agency for SMEs, TIPA. The funders had no role in study design, data collection and analysis, decision to publish, or preparation of the manuscript.

**Competing interests:** The authors have declared that no competing interests exist.

## Introduction

Globally, the population is aging rapidly. According to the United Nations, the proportion of people aged 65 years and above is projected to double approximately in the next 25 years. Aging is a major social, economic, and cultural concern [1]. As men age, the amount of testosterone produced declines, leading to physical and mental changes, such as reduced libido, reduced sperm production and activity, reduced muscle strength, reduced body hair, depression, lethargy, and osteoporosis. These changes are collectively referred to as testosterone-deficiency syndrome (TDS) [2, 3]. Low testosterone levels are also associated with an increased prevalence of metabolic and cardiovascular disease [4–7]. However, these changes do not always occur simultaneously. One or more of these changes are typically experienced in conjunction with low testosterone levels.

Several studies have implicated mitochondrial dysfunction as the molecular cause of TDS with reduced steroidogenesis [8, 9]. Mitochondria are the centers of cellular energy metabolism, performing both catabolic and anabolic processes [10], and thus determining the quality and rate of androgen formation [8]. Mitochondria are also both a major source of ROS and targets of ROS damage. With age, the cell's ability to scavenge ROS declines, and ROS overproduction eventually causes mitochondrial dysfunction [11, 12]. TDS is a global concern due to its association with age. Testosterone replacement therapy (TRT) is the mainstay of TDS treatment. Clinically, the primary goal of TRT is to restore normal testosterone levels in response to testosterone deficiency, rather than to elevate testosterone levels. Testosterone supplements used for TRT are available in several different formulations [13, 14]. However, the drugs themselves are steroid hormones and carry the potential risk of side effects [15, 16]. Therefore, research and development of TDS derived from natural products that are safe from side effects are urgently needed. Recently, alternative studies have been conducted to develop new substances that boost testosterone. These substances promote testosterone secretion by regulating the body's production of testosterone by using naturally occurring materials, such as plants, fungi, and insects [17–20]. In addition, further information on the bioavailability and efficacy of the rich bioactive compounds in natural products is needed, due to the growing public interest in the development of "functional foods" [21, 22] and the increasing trend for health-consciousness among consumers [23, 24].

*Sambucus nigra* L. is commonly known as 'elderberry', 'black elderberry' and 'European elderberry'; it is part of the family Adoxaceae. Elderberries contain many bioactive compounds, mainly anthocyanin derivatives, including cyanidin 3-glucoside, cyanidin 3-sambubioside, cyanidin 3-rutinoside, cyanidin 3-sambubioside-5-glucoside, and cyanidin 3,5-diglucoside, with cyanidin 3-sambubioside or cyanidin 3-glucoside being the most abundant depending on the variety and ripeness. At present, anthocyanins are becoming very popular because they are potential natural alternatives to synthetic food colourants, and, in addition, their antioxidant capacity plays an important role in the prevention of different diseases. In our previous research on the components of elderberry extract, we identified rutin as a indicator and fructose-leucine (FL) as a candidate for functional effects. Our results showed that rutin is the most abundant and stable component. Rutin content was 2.74 mg/g. Elderberry supports the immune system [25, 26], respiratory [27, 28] and cardiovascular health [29], and has anti-inflammatory effects [30]. Recent studies have suggested that it is a potent antioxidant [31–33]. However, research on using elderberries and indicator substances for diseases related to reduced testosterone levels, such as sarcopenia and older age, is at an early stage. In our previous studies, we analyzed the components of a water extract of elderberry and identified a monosaccharide-amino acid (fructose–leucine; FL) as a candidate substance that may improve testosterone secretion in TDS. FL is not simply a combination of fructose

and amino acids; the parent sugar of FL is glucose, and FL is the product of an *Amadori* rearrangement of N-glycosylamino acids [34]. However, the chemistry of FL and how FL works is poorly understood. In this study, we identified an indicator compound in elderberry extracts. Subsequently, TM3 (mouse Leydig) cells, treated with $H_2O_2$ to decrease testosterone biosynthesis, were used to test the effects of elderberry extract and candidate extract-derived compounds on the mechanism of testosterone biosynthesis.

## Materials and methods

### Preparation of elderberry extract

Elderberry extract was prepared using 10 times the weight of elderberries in purified water. The elderberry has been extracted once at a temperature of 80°C for 3 h. After being filtered (60 mesh) and concentrated (15 Brix°), 30% of the concentrated extract was mixed with dextrin and freeze dried to produce the powder. All the processes were performed in a GMP production facility (Samwoodayeon lnc., Geumsan-gun, Chungcheongnam-do, Korea) (S1–S6 Tables and S1–S3 Figs). The powdered, freeze-dried extract was stored at -70°C until used in the experiment.

### Cell culture

The TM3 cells used in the experiments were purchased from the American Type Culture Collection (Manassas, VA, USA). Cells were cultured using a 1:1 mixture of Dulbecco's modified Eagle's medium (Gibco, Waltham, MA, USA) and F-12K medium, to which 5% fetal bovine serum (Merck Millipore, Berlin, Germany), 5% horse serum (Welgene, Seoul, Korea), and 1% antibiotics (penicillin/streptomycin, Gibco) were added. Cells were cultured in a cell incubator (INS 153, Memert, Schwabach, Germany) at 95% humidity, 5% $CO_2$, and 37°C.

### Cell viability assay

TM3 cells were seeded in 96-well culture plates at $5 \times 10^4$ cells/well in 90 μL. The cells were treated with elderberry extract and Fructose-leucine (FL) (SC-470657A, Santa Cruze Biotechnology Inc., TX, USA, Lot I0123, CAS 34393-18-5) for 24 and 48 h. Elderberry extract was dissolved in water as the solvent for a stock solution and then diluted to concentrations of 15.63, 31.25, 62.5, 125, 250, and 500 μg/mL for treatment of each well. FL was also treated with water as a solvent to provide a fully dissolved, sediment-free stock solution and subsequently diluted to concentrations of 7.81, 15.63, 31.25, 62.5, 125, 250, and 500 μg/mL. The same amount of solvent was included in the control. Subsequently, 10 μL MTS reagent (EZ-cytox, Doogen, Seoul, Korea) was added to each well and absorbance was measured at 450 nm after 4 h. Cell viability was expressed relative to untreated elderberry extract and FL.

### Intracellular reactive oxygen species

The TM3 cells were seeded at a density of $5 \times 10^4$ cells/well in 96-well culture plates in 90 μL and incubated overnight for 24 h. Treatment with $H_2O_2$ (600 μM, Sigma-Aldrich, St. Louis, USA) was used to down-regulate testosterone biosynthesis, and the experimental method was adapted and modified from that used by Greifová et al [35]. Elderberry extract and FL were added the culture medium at different concentrations and the cells further incubated for 24 h. They were then treated with 2',7'-dichlorofluorescin diacetate. Fluorescence values at 480/520 nm were measured after 30 min. The ROS levels were analyzed in triplicate for each treatment and the measurements were compared to the untreated control.

## Testosterone secretion

TM3 cells were seeded in 24-well culture plates at $1\times10^5$ cells/well in 500 μL and allowed to stabilize for 24 h. The medium was then replaced with serum-free medium, and elderberry extract and FL were diluted and added to each well. The culture medium was collected after 24 h. The collected medium was centrifuged to obtain the supernatant, which was analyzed using a testosterone ELISA kit (Cayman Chemical, Ann Arbor, MI, USA). Reagents were added according to the manufacturer's instructions and incubated for 2 h at room temperature after 50 μL of cell supernatant was added to each well. The assay was then washed twice. The absorbance was measured at 405 nm 1 h after the addition of the chromogenic reagent. For quantitative analysis of testosterone, the concentration of each sample was determined using the equation from the standard curve.

## Quantitative real-time reverse-transcription polymerase chain reaction

TM3 cells were seeded at $5\times10^5$ cells/well in 2 mL in 6-well culture plates and incubated overnight. They were then treated with $H_2O_2$ (600 μM) and elderberry extract and FL at appropriate concentrations and incubated for 24 h. RNA extraction was performed according to the manufacturer's instructions using the Minibest Universal RNA Extraction Kit (Takara, Shiga, Japan). The extracted RNA was quantified (BioSpec-nano, SHIMAZU, Kyoto, Japan), and cDNA synthesis was performed by adding 2 μg of total RNA to cDNA premix (TaKaRa, RNA to cDNA EcoDry Premix) and adjusting the final volume to 20 μL, followed by 60 min at 42˚C, 10 min at 70˚C, and a cooling step. The synthesized cDNA was then diluted 5-fold and used for quantitative reverse transcription-polymerase chain reaction (RT-PCR) of steroidogenic acute regulatory protein (*star*), cytochrome P450 11A1(*cyp11a1*), cytochrome P450 17A1(*cyp17a1*), cytochrome P450 19A1(*cyp19a1*, aromatase), 3β-hydroxysteroid dehydrogenase *(3β-had)*, 17β-hydroxysteroid dehydrogenase *(17β-hsd)*, and *β-actin* genes (Light Cycler 2.0, Roche, Mannheim, Germany). DNA was first denatured at 95˚C for 30 sec, followed by annealing at 60˚C for 30 sec and extension at 72˚C for 30 sec. This was repeated for 35 cycles before a final extension step at 72˚C for 10 min. The β-actin gene was used as an internal control for mRNA normalization. Fluorescence signals were normalized to that of the internal reference, and the quantification cycle (Cq) was set within the exponential phase of the PCR. Relative mRNA expression levels were calculated using the $2^{-(\Delta Ct\ sample-\Delta Ct\ control)}$ method. Each primer was sequenced as follows (Table 1):

**Table 1. Primer sequences for quantitative reverse-transcription polymerase chain reaction (RT-PCR) analysis.**

| Gene | | Primer sequence (5'–3') | Amplicon length (bp) |
|---|---|---|---|
| *Star* | Forward | GGAAGTCCCTCCAAGACTAAAC | 281 |
| | Reverse | AGTCCTAGTGTCTCCTGACTAC | |
| *3β-hsd* | Forward | GTAACAGTGTTGGAAGGAGACA | 99 |
| | Reverse | GACATCAATGACAGCAGCAGTG | |
| *17β-hsd* | Forward | TTGTTTGGGCCGCTAGAAG | 43 |
| | Reverse | CACCCACAGCGTTCAATTCA | |
| *Cyp11a1* | Forward | GTCCTTCAATGAGATCCCTTCC | 277 |
| | Reverse | CCCAATGGGCCTCTGATAATAC | |
| *Cyp17a1* | Forward | CTCCAGCCTGACAGACATTCTG | 117 |
| | Reverse | TCTCCCACCGTGACAAGGAT | |
| *Cyp19a1* | Forward | TGGAAAACAACTCGACCCTTCT | 73 |
| | Reverse | CACAGACTGTGACCATACGAACAA | |
| *β-actin* | Forward | AGAGAAGCTGTGCTATGTT | 179 |
| | Reverse | CACAGGATTCCATACCCAAG | |

## Western blot

The TM3 cells were seeded at $1 \times 10^6$ cells per well in 100 mm culture dishes and cultured overnight. The cells were treated with $H_2O_2$ (600 μM) and elderberry extract and FL at the indicated concentrations and incubated for 24 h. The culture medium was then removed. Cell culture dishes were washed twice with cold $1 \times$ phosphate buffered saline (PBS) and proteins were extracted using lysis buffer (Cell Signaling Technology Inc., Danvers, MA, USA). The concentration of the extracted proteins were analyzed using a Bradford assay and equal amounts were separated by sodium dodecyl sulfate polyacrylamide gel electrophoresis. Proteins were transferred onto a polyvinylidene fluoride membrane (PVDF membrane, Bio-Rad, Hercules, CA, USA). A blocking buffer containing 5% skim milk powder in 1XTBS-T solution was then used to block the non-specific binding of proteins. Blocked for 1 hour at room temperature. After washing of the membrane, the membrane was incubated with antibodies against star (Santa Cruz Biotechnology, Santa Cruz, CA, USA, 1:1000), cyp11a1 (Abcam, Cambridge, UK, 1:1000), cyp17a1 (Abcam, 1:1000), 3β-hsd (Santa Cruz Biotechnology, 1:1000), 17β-hsd (Santa Cruz Biotechnology, 1:1000), cyp19a1(Invitrogen, 1:500) and β-actin (Santa Cruz Biotechnology, 1:2000), overnight at 4˚C. The next day, the membrane was washed with $1 \times$ TBS-T buffer and was then incubated with the secondary antibody (anti-mouse, anti-rabbit, Santa Cruz Biotechnology) for 1 h at room temperature. Secondary antibody to β-actin was incubated at 1:4000 concentration and other antibodies including star at 1:2000 concentration at room temperature (S7 Table). The membrane was again washed in $1 \times$ TBS-T, after which protein expression was detected using enhanced chemiluminescence western blotting substrate (Promega, Madison, WI, USA). The blot images were analyzed using the ChemiDoc™ XRS+ Imaging System (Bio-Rad).

## Statistical analysis

The data are expressed as mean ± SD. Student's t-test or one-way analysis of variance (ANOVA) in SPSS version 22 (SPSS Inc., Chicago, IL, USA) with Tukey's post hoc test was used for the analysis of differences between groups. The level of significance was considered as $^*p{<}0.05$, $^{**}p{<}0.01$, $^{***}p{<}0.001$.

## Results

### Cell viability related to elderberry extract and FL treatment

In previous our studies, FL, an active indicator component derived from elderberry extract believed to have a beneficial effect on low testosterone secretion, was analyzed. FL is a six-carbon monosaccharide containing the amino acid leucine (MW 293.31) (S1 and S2 Figs). FL is not simply a combination of fructose and amino acids; the parent sugar of FL is glucose, and FL is the product of an *Amadori* rearrangement of N-glycosylamino acids [34]. To determine the effect of elderberry extract and FL on TM3 cells viability, we determined cell viability at concentrations from 0 to 500 μg/mL (Fig 1). Elderberry extract from 0 to 62.5 μg/mL showed no effect on cell viability, but viability was reduced as the concentration increased thereafter, with 80% viability observed at a treatment concentration of 250 μg/mL and 70% cell viability observed at a treatment concentration of 500 μg/mL (Fig 1(A)).

The viability of Leydig cells after 48 h of treatment with elderberry extract was similar to that after 24 h of treatment. Based on the results of cell viability at 24 and 48 h, the dose of 250 μg/mL was not exceeded in the subsequent experiments (Fig 1(B)). On the other hand, FL did not affect Leydig cell viability at either 24 or 48 h, even at the highest treated concentration of 500 μg/mL (Fig 1(C, 1D)). In our previous study on the content of FL in elderberry extract,

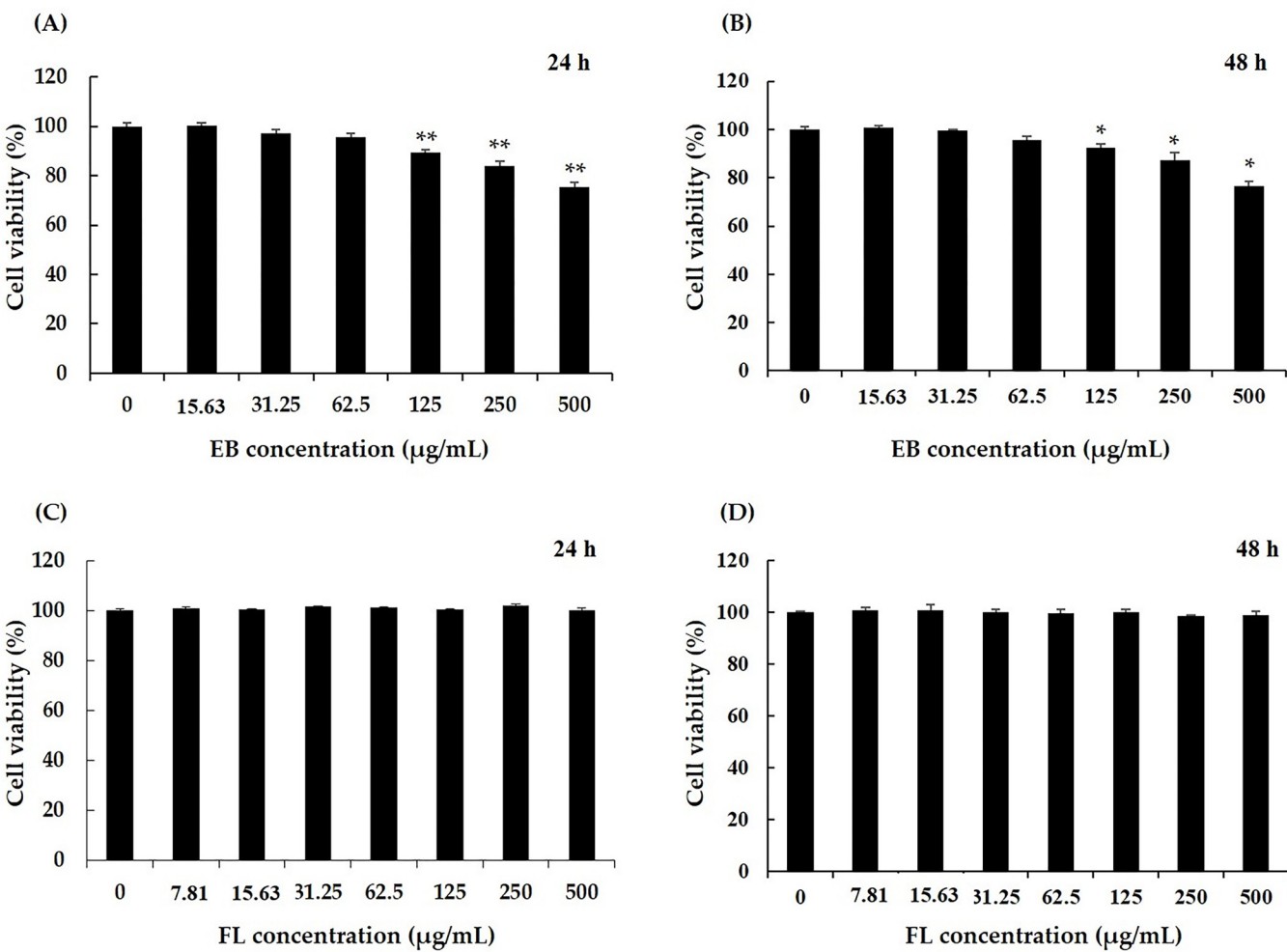

**Fig 1. Effect of elderberry extract and extract-derived FL compound.** TM3 cells were treated with different concentrations of EB (A) and FL (B) for 24 and 48 h before an MTS assay was performed. Data are expressed as mean ± SD (n = 3). Student's t-test was used to analyze differences between groups at *$p < 0.05$, **$p < 0.01$. EB; elderberry extract, FL; fructose–leucine.

the FL component was confirmed as 0.5% (S6 Table and S2 and S3 Figs), so the concentration of FL was calculated from the treatment concentration of elderberry extract to perform the cell assay. That is, the cell treatment capacity of the calculated FL was set from 5 μg/mL to 20 μg/mL.

## Induction of reduction of testosterone production in TM3 cells with $H_2O_2$

In this study, we treated TM3 cells with $H_2O_2$ to induce a decrease in testosterone biosynthesis [36–38] and then determined the changes in cell viability, cytotoxicity, testosterone secretion, and *star* mRNA expression (Fig 2). In this experiment, 300, 600, and 900 μM $H_2O_2$ were used for treatment. As the concentration of $H_2O_2$ increased, the viability of the TM3 cells decreased (Fig 2(A)). The lactate dehydrogenase (LDH) level, which was analyzed as an indicator of cytotoxicity, increased accordingly (Fig 2(B)). At the 600 μM treatment concentration, cell viability reduced by approximately 20%, and at the 900 μM concentration, cell viability reduced by about 40%. Testosterone secretion by TM3 cells was reduced following $H_2O_2$ treatment (Fig 2(C)). A concentration-dependent decrease in testosterone secretion was observed in all treatment groups as the $H_2O_2$ concentration increased. These results were consistent with the

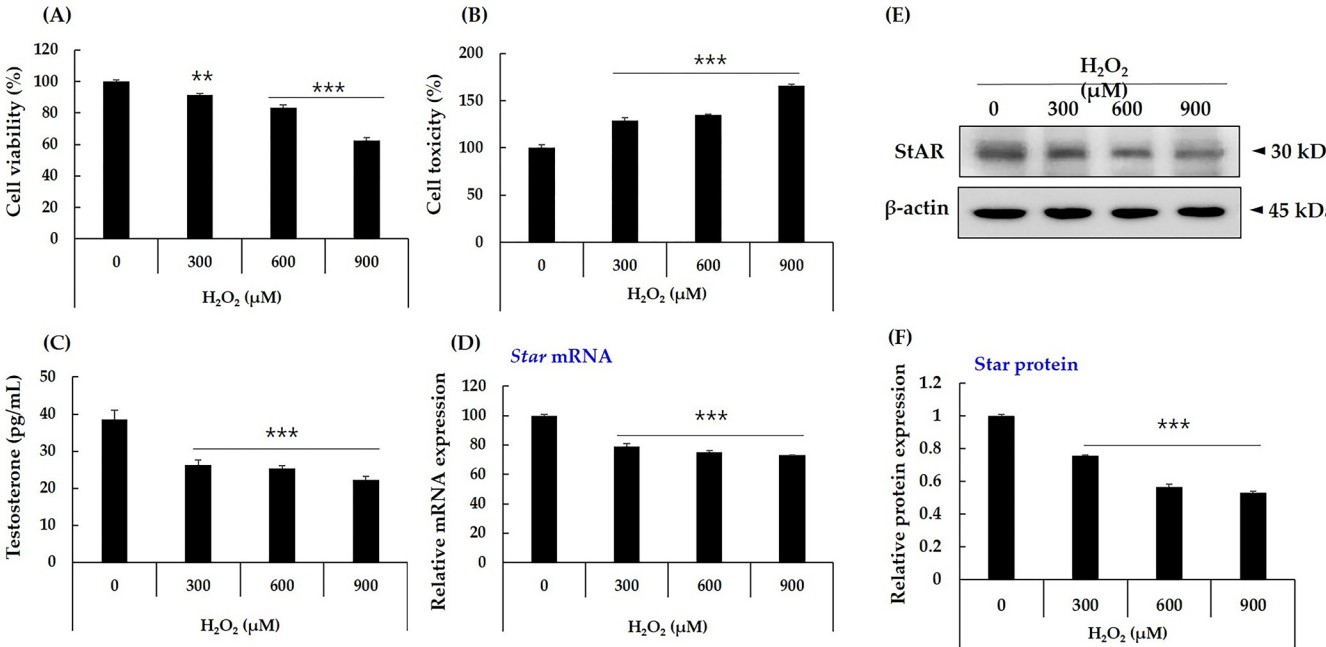

**Fig 2. Induction of a reduced production of testosterone in TM3 cells by treatment with H₂O₂.** TM3 cells were treated with H₂O₂ (600 μM) for 24 h and analyzed for viability (A), cytotoxicity (B), testosterone secretion (C), StAR mRNA (D) and protein expression (E). Data are expressed as mean ± SD (n = 3). Student's t-test was used to analyze differences between groups at **$p<0.01$. ***$p<0.001$.

observed *star* mRNA and protein expression, indicating that H₂O₂ treatment caused decreased star protein and *star* mRNA expression, which ultimately led to decreased testosterone secretion (Fig 2(D–2F)).

## Ameliorative effects of elderberry extract and FL on H₂O₂-induced decrease in testosterone biosynthesis

**Intracellular ROS.** Many factors can lead to a decrease in testosterone synthesis in the testes, including aging, stress, and the side effects of medications. In the present study, we used H₂O₂ to induce oxidative stress in TM3 cells to create a cellular model of decreased testosterone production (Fig 2). Ultimately, the production of intracellular ROS was increased in H₂O₂-treated TM3 cells. As the elderberry extract treatment concentration increased, the increased intracellular ROS levels significantly decreased (Fig 3(A)). In particular, the group treated with elderberry extract at a concentration of 200 μg/mL showed a 40% inhibition of ROS as compared to the H₂O₂-treated group. In this study, we determined the effect of FL on the production of ROS induced by H₂O₂ (Fig 3(B)). The H₂O₂-induced ROS levels were significantly reduced by treatment with FL and showed a decreasing trend in a concentration-dependent manner. ROS are an important factor in mitochondrial dysfunction in Leydig cells. As the resistance of cells decreases with age, intracellular organelles are affected by intracellularly generated ROS. Mitochondria, which are responsible for anabolism and catabolism through the electron transport system, are particularly susceptible to ROS. In addition to the electron transport system, mitochondria contain enzymes involved in testosterone biosynthesis, and these enzymes can be affected by ROS, which has implications for their functional role. This in turn leads to a decrease in the biosynthesis of testosterone. Therefore, a key regulator of testosterone production is the scavenging of intracellular ROS. Many studies, including Ferreira et al. (2022), have reported intracellular ROS scavenging by elderberry extract

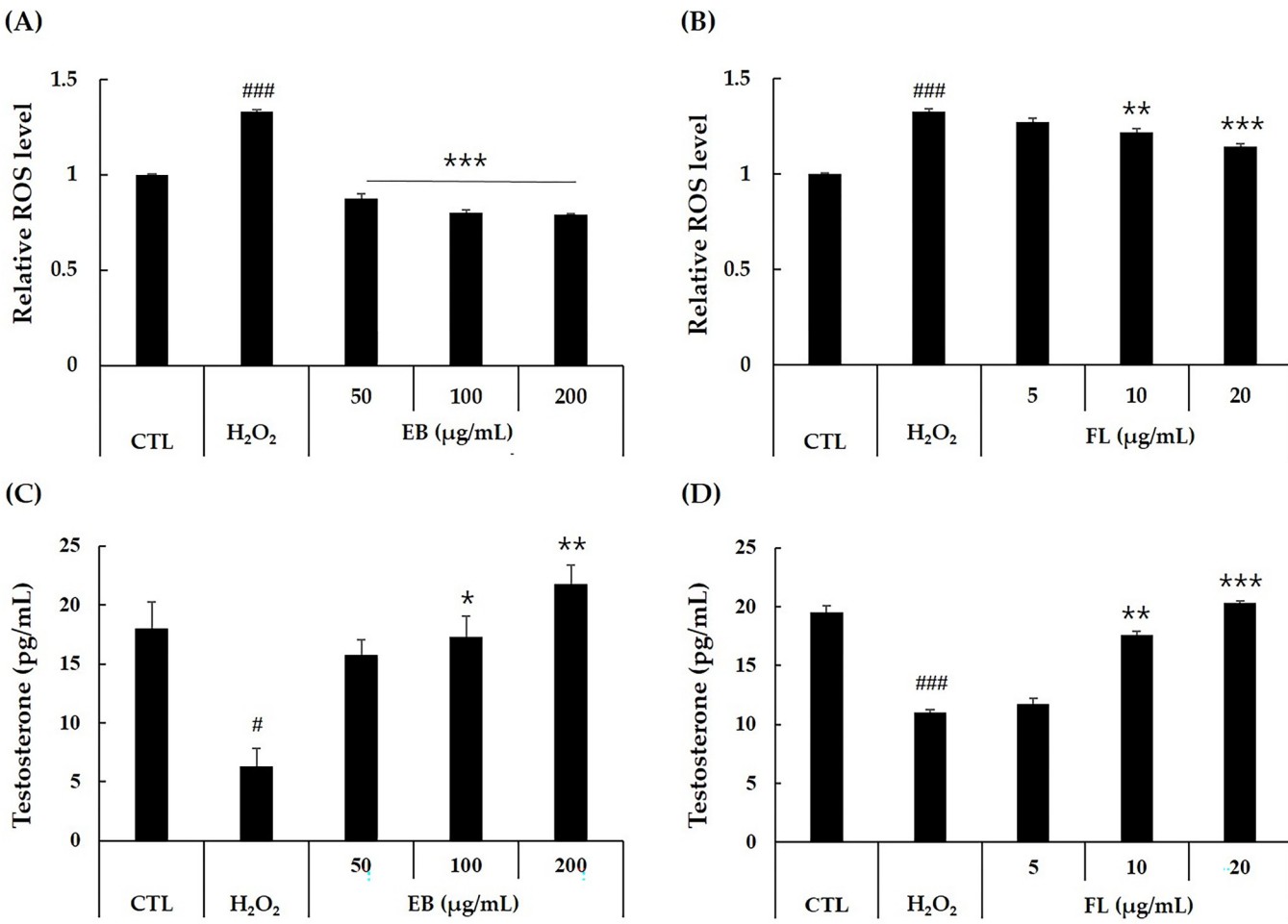

**Fig 3. Effects of elderberry extract and elderberry extract-derived FL on inhibition of $H_2O_2$ treatment-induced reactive oxygen species (ROS) production and improvement of testosterone secretion in TM3 cells.** TM3 cells were treated with EB and FL combined with $H_2O_2$ (600 μM) for 24h and analyzed for intracellular ROS (A, B) and testosterone secretion (C, D). Data are expressed as mean ± SD (n = 3). [#]$p < 0.05$, [###]$p < 0.001$ versus the control group, $p < 0.05$, [**]$p < 0.01$, [***]$p < 0.001$ versus the treated with $H_2O_2$ group, analyzed via one-way ANOVA. CTL: control, EB: elderberry extract, FL: fructose–leucine.

[39–41]. Scavenging of intracellular ROS by elderberry extract has been reported in many research papers Ferreira et al. (2022) [39–41]. On the other hand, our study is about the ROS inhibitory effect of FL, a monosaccharide amino acid, which is not an anthocyanin component.

**Testosterone secretion.** Elderberry treatment also reduced testosterone secretion after $H_2O_2$ treatment (Fig 3(C)). Specifically, in this experiment, treatment with elderberry extract at a concentration of 100 μg/mL improved testosterone secretion to normal levels. FL treatment also significantly increased the $H_2O_2$-reduced testosterone secretion, similar to elderberry extract treatment (Fig 3(D)). In particular, the group treated with FL at a concentration of 10 μg/mL showed improvement to normal levels. In addition to our studies, many other herbs have been studied for their ability to boost testosterone and improve TDS, including *Trigonella foenum-graecum* (Fenugreek), *Withania somnifera* (Ashwagandha), *Tribulus terrestris* (Tribulus), *Lepidium meyenii* (Maca), *Rhodiola rosea* (Rhodiola), *Chlorophytum borivilianum* (Musali), *Garcinia cambogia* (Garcinia), *Coleus forskohlii* (Forskohlii), *Panax ginseng* (Asian Ginseng), and others [29]. However, only fenugreek seed extract (*Trigonella foenum-graecum*) and ashwagandha roots and leaves (*Withania somnifera*; water-based or ethanol:water, 70:30,

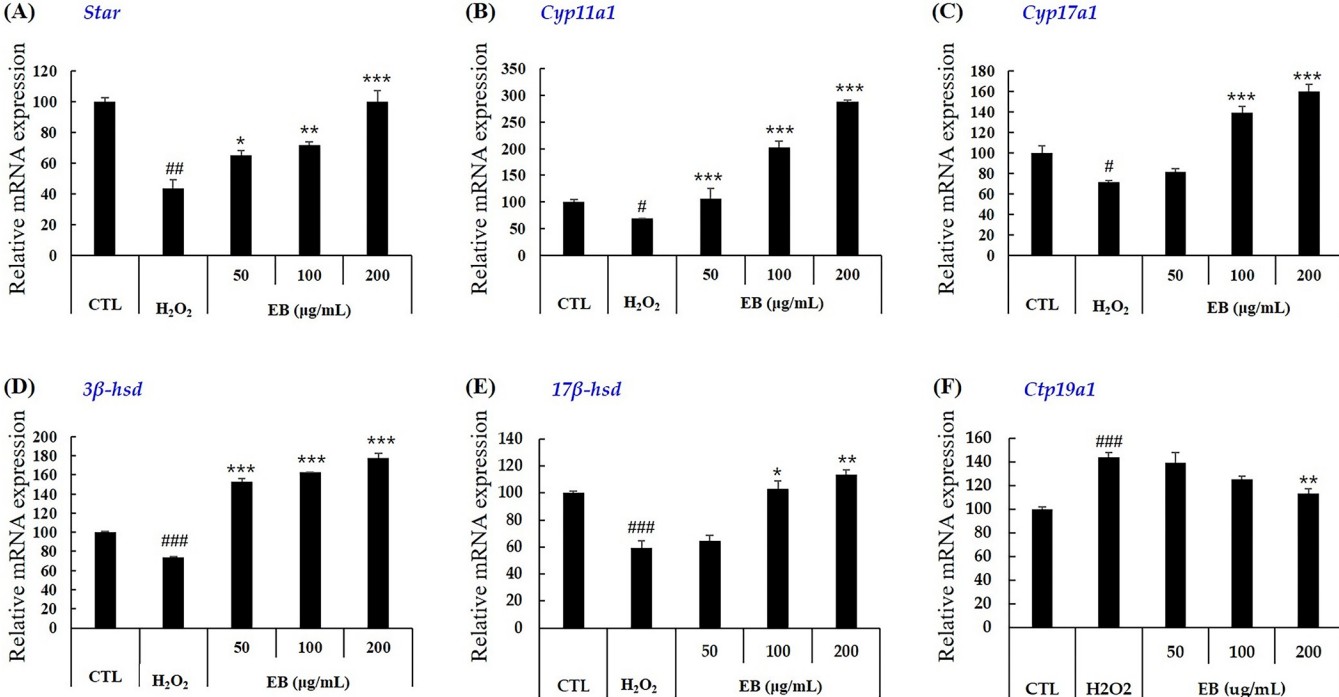

**Fig 4. Effect of elderberry extract on expression of genes encoding enzymes involved in steroidogenesis in TM3 cells treated with $H_2O_2$.** The mRNA expression of genes encoding enzymes involved in steroidogenesis was analyzed after these cells were treated with EB and $H_2O_2$ (600 μM) for 24 h. The expression was determined by quantitative real-time reverse -transcription polymerase chain reaction. Data are presented as mean ± SD (n = 3). Data are expressed as mean ± SD (n = 3). $^#p < 0.05$, $^{##}p < 0.01$, $^{###}p < 0.001$ versus the control group, $^*p < 0.05$, $^{**}p < 0.01$, $^{***}p < 0.001$ versus the treated with $H_2O_2$ group, analyzed via one-way ANOVA. CTL: control, EB: elderberry extract, FL: fructose–leucine.

extracts) have shown consistent efficacy among the numerous studies on testosterone biosynthesis. To draw definitive conclusions about the testosterone biosynthetic effects of other natural botanicals, high-quality studies of the herbs used must be carefully conducted.

**Gene expression.** Gene expression analysis of enzymes involved in testosterone biosynthesis was performed based on the increased secretion of testosterone (Fig 4). Elderberry extract treatment increased mRNA levels of *star*, *cyp11a1*, *cyp17a1*, *3β-hsd* and *17β-hsd* genes involved in testosterone biosynthesis, which had been decreased by $H_2O_2$ (Fig 4(A–4E)). We also examined changes in the expression of *cyp19a1* (aromatase), which converts testosterone to estradiol (Fig 4(F)). Elderberry extract decreased *cyp19a1* expression. Elderberry extract treatment affected gene expression slightly differently, but most genes showed a return to normal mRNA levels or a 1.5-3fold increase. Furthermore, the expression of testosterone biosynthetic enzymes (*star*, *cyp11a1*, *cyp17a1*, *3β-hsd* and *17β-hsd*), which were reduced by $H_2O_2$ treatment, were restored by FL treatment (Fig 5(A–5E)). In addition, FL treatment inhibited the expression of an enzyme that converts testosterone to estradiol by inhibiting *cyp19a1*, confirming the results obtained with the elderberry extract (Fig 5(F)). These results suggest that elderberry extract and FL may increase the biosynthesis and secretion of testosterone by increasing the expression of genes involved in the biosynthesis of testosterone, which was reduced by the treatment with $H_2O_2$. They also inhibited the conversion of male to female hormones by reducing aromatase at the mRNA level.

**Protein expression.** The gene expression results were identical to the protein expression results for enzymes involved in testosterone biosynthesis. In other words, the protein expression of testosterone biosynthetic enzymes reduced by $H_2O_2$ was increased by elderberry

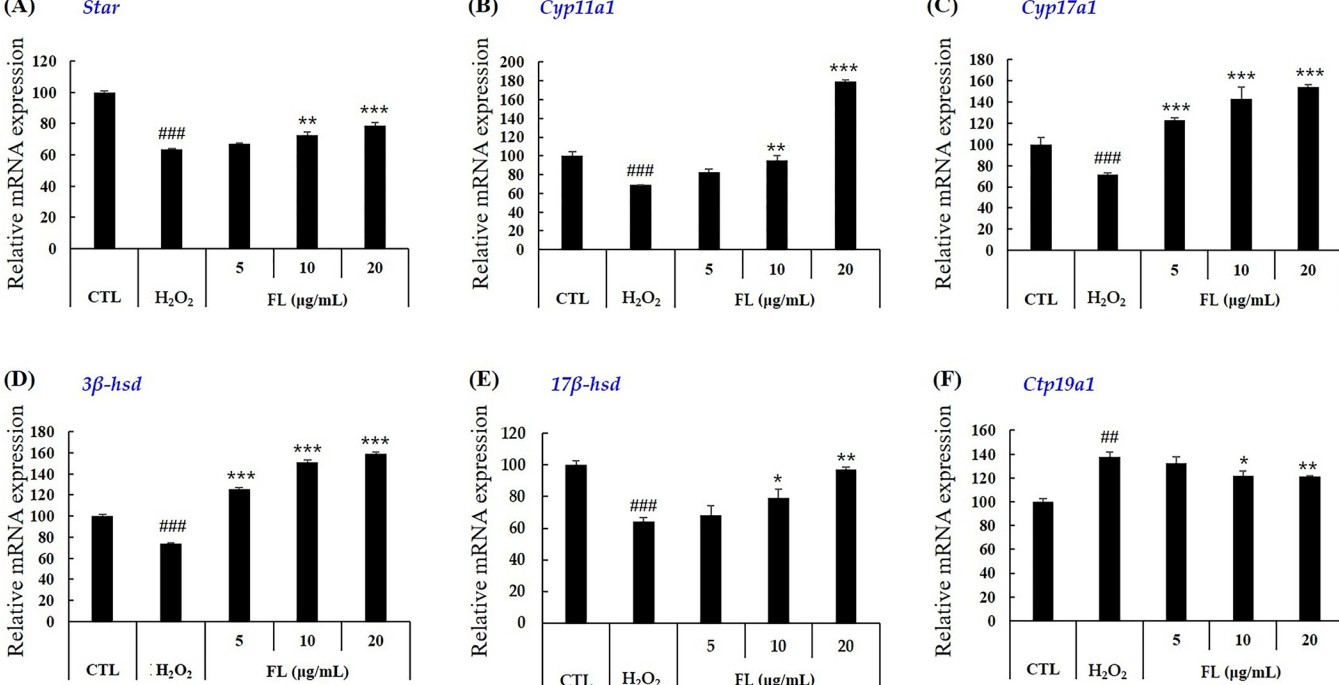

**Fig 5. Effect of FL derived from elderberry extract on the expression of genes encoding enzymes involved in steroidogenesis in TM3 cells treated with H₂O₂.** The mRNA expression of these genes was analyzed after TM3 cells were treated with FL and $H_2O_2$ (600 μM) for 24 h. The expression was determined by quantitative real-time reverse-transcription polymerase chain reaction. Data are presented as mean ± SD (n = 3). Data are expressed as mean ± SD (n = 3). ##$p < 0.01$, ###$p < 0.001$ versus the control group, $p < 0.05$, **$p < 0.01$, ***$p < 0.001$ versus the treated with $H_2O_2$ group, analyzed via one-way ANOVA. CTL: control, EB: elderberry extract, FL: fructose–leucine.

extract and FL treatments (Fig 6). Specifically, elderberry extract and FL treatment increased protein expression levels of testosterone biosynthetic enzymes, star, cyp11a1, cyp17a1, 3β-hsd and 17β-hsd, reduced by $H_2O_2$, to normal levels or 1.2–2 times normal levels (Fig 6(A–6F)). However, cyp19a1 expression showed the opposite result. The $H_2O_2$-treated group showed increased cyp19a1 protein expression, which was predictably supported by *cyp19a1* mRNA expression. Furthermore, the protein expression of cyp19a1 was decreased in a concentration-dependent manner in the elderberry extract and extract-derived FL treatment group compared to the $H_2O_2$ treatment group (Fig 6(G)). These results predict that by inhibiting not only the mRNA expression but also the protein expression of *cyp19a1*, elderberry extract and extract-derived FL reduce the conversion of testosterone to estradiol. These results suggest that by modulating the expression of mRNA and proteins involved in steroidogenic enzymes, elderberry extract and elderberry extract-derived FL may contribute to the regulation of testosterone production and secretion.

## Discussion

Humans strive to maintain their health by consuming foods and derived functional foods that are rich in nutritional and functional value, to counteract various diseases that occur during the aging process. Among them, elderberry (*Sambucus nigra* L.) is well known for its ability to support immune, respiratory, and cardiovascular health. It is also known for its anti-inflammatory properties. In addition, recent research has shown that it provides many health benefits to the human body as a powerful antioxidant. In addition to the benefits mentioned above, studies have reported improvements in the metabolic syndrome, antiviral, antibacterial and

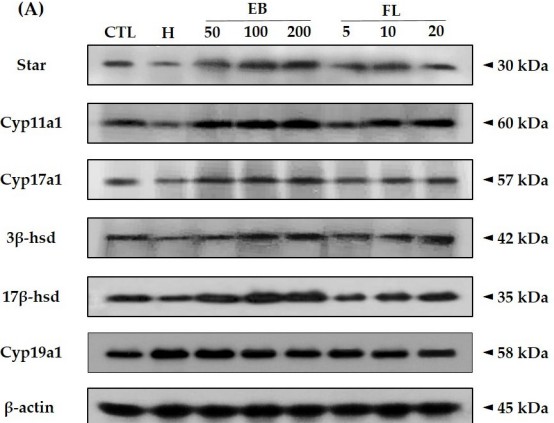

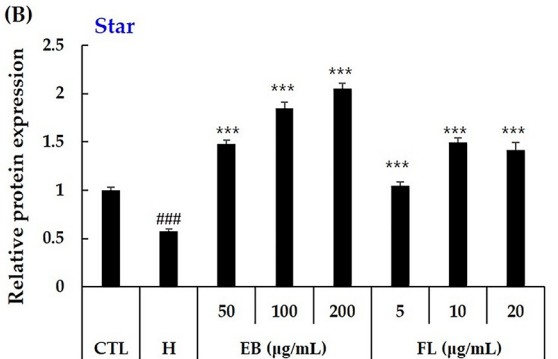

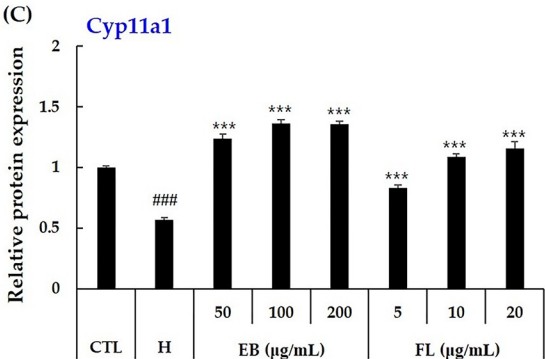

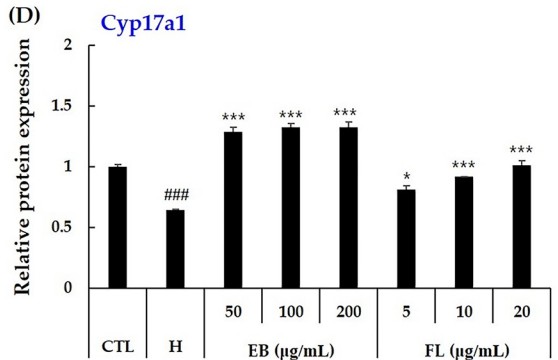

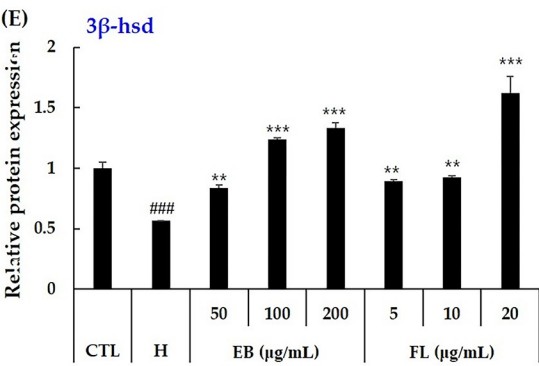

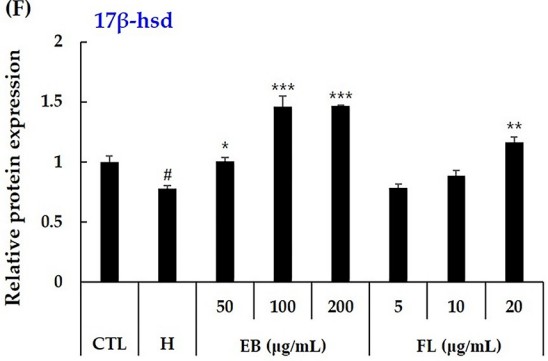

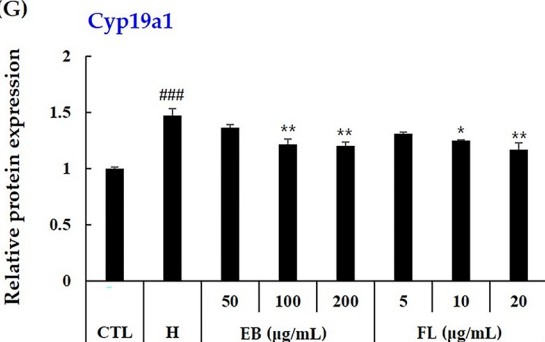

**Fig 6. Effect of elderberry extract and elderberry extract-derived FL on the increase in protein expression of steroidogenesis-related enzymes in TM3 cells treated with $H_2O_2$.** The protein expression of the enzymes involved in steroidogenesis in TM3 cells treated with EB and FL combined with $H_2O_2$ (600 μM) for 24 h was determined by western blotting. Data are presented as mean ± SD (n = 3). Data are expressed as mean ± SD (n = 3). [#]$p < 0.05$, [###]$p < 0.001$ versus the control group, $p < 0.05$, [**]$p < 0.01$, [***]$p < 0.001$ versus the treated with $H_2O_2$ group, analyzed via one-way ANOVA. CTL: control, EB: elderberry extract, FL: fructose–leucine.

antifungal effects, anti-cancer effects, and protection of the skin from UV radiation [17, 42–47].

In this study, we investigated the effects of elderberry extract and extract-derived FL on the gene and protein expression of enzymes involved in testosterone secretion and steroidogenesis. For this experiment, we used Leydig cells that were derived from the testes of a mouse. The TM3 cells are stromal cells in the testes responsible for testosterone production [48]. Leydig cells are affected by various stimuli such as aging, disease, stress and inflammation [49, 50]. This leads to Leydig cell dysfunction. The dysfunction of the Leydig cells is a direct contributor to the problem of low testosterone secretion. Therefore, we treated Leydig cells with $H_2O_2$ to induce intracellular oxidative stress in order to induce functional defects in the Leydig cells. In $H_2O_2$-treated Leydig cells, ROS were generated, and it was found that the generated ROS decreased the secretion of testosterone.

Testosterone is a hormone of the steroid family that is produced by Leydig cells, interstitial cells in the seminiferous tubules of the testes. It is secreted into the bloodstream where it circulates throughout the body and acts on many parts of the body. Testosterone is the hormone of masculinity and plays an important role in muscle and skeletal development, increased body hair, testicular and prostate development, and normal sperm production and motility [51–57]. Women also produce small amounts of testosterone in the ovaries and adrenal glands, and androgens in women, as in men, are essential for the maintenance of reproductive competence, cardiac health, proper bone remodeling and mass maintenance, muscle tone and mass, and brain function, in part by mitigating the effects of neurodegenerative disease [58–62].

Testosterone production is stimulated by the binding of luteinizing hormone (LH) to its receptor, the LH receptor (LH-R). This activates the adenylyl cyclase and increases the production of cAMP and cAMP-dependent protein kinase [63]. This signal activates StAR and translocator protein (TSPO) on the inner mitochondrial membrane, which is a critical step in initiating testis steroidogenesis [64, 65]. Cholesterol is used as a substrate for steroid hormone production, metabolized by the cytochrome P450 cholesterol-scavenging enzyme (P450cc) to pregnenolone, which is then converted to testosterone via a chain reaction involving 3β-HSD, cytochrome P450 17α-hydroxylase/C17-20 lyase (P450c17), and 17β-HSD [66, 67]. In this context, the testosterone biosynthetic capacity of Leydig cell is strongly associated with cellular aging and mitochondrial dysfunction [68]. Mitochondria are organelles that generate more than 90% of ROS in their role as regulatory centers of energy metabolism [69, 70]. Consequently, they are also target of ROS damage. Although normal cells scavenge ROS in various ways, aging cells are less capable of doing so. Excess ROS eventually causes damage to organelles, including mitochondria. Sustained accumulation of such damage leads to mitochondrial dysfunction, particularly in Leydig cells. This in turn leads to decreased steroidogenesis and ultimately contributes to the development of TDS [71].

It has been well known for over 30 years that there is a strong correlation between low testosterone levels and a variety of diseases. Specifically, low testosterone has been reported to increase the risk of coronary artery disease (CAD), metabolic syndrome, and type 2 diabetes [72–75]. Therefore, our work on elderberry extracts and extract-derived FL is expected to be an interesting research topic not only for improving TDS, but also for treating other age-related diseases. $H_2O_2$ caused ROS production in TM3 cells. ROS significantly decreased

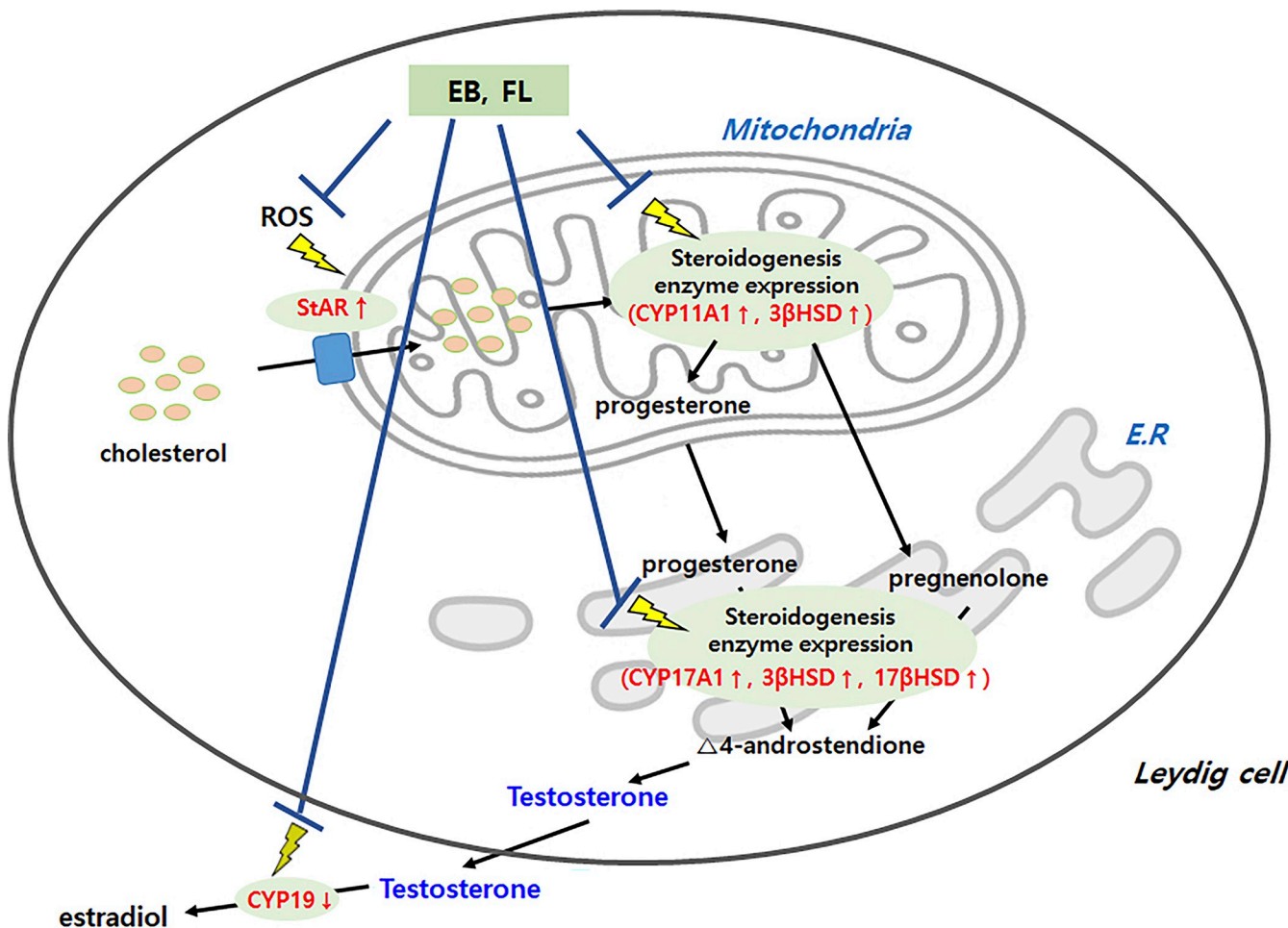

**Fig 7. Schematic representation of the effects of elderberry extract and the monosaccharide-amino acids derived from elderberry extract in the improvement of testosterone-deficiency syndrome.** EB: elderberry extract, FL: Fructose–leucine, E·R: endoplasmic reticulum, ROS: reactive oxygen species.

mRNA and protein expression of StAR, CYP11A1, CYP17A1, 3β-HSD, and 17β-HSD, genes involved in steroid biosynthesis. There are several known causes of Leydig cell dysfunction, including disease, inflammation, stress, and aging, which result in cellular changes, such as decreased leukocyte numbers, formation of small intracellular clusters, cell atrophy, and multi-nucleation [68, 69]. These cellular changes ultimately contribute to a decrease in testosterone biosynthesis and secretion in the Leydig cells. Therefore, the present study confirmed that an in vitro model for decreased testosterone secretion following $H_2O_2$ treatment was appropriately derived and that $H_2O_2$ affects star protein expression and the regulation of testosterone secretion in Leydig cells, as reported by Tsai *et al.* [36]. A decrease in testosterone secretion by $H_2O_2$ (600 μM) treatment was also reported by Greifová et al [35].

In our previous study, we analyzed the composition of elderberry extracts, isolated and purified potentially active compounds that could help improve steroidogenesis, and identified FL. As noted above, FL is not simply a combination of fructose and an amino acid; the parent sugar of FL is glucose, and FL is one of the products of the *Amadori* conversion of an N-glycosyl amino acid. The *Amadori* rearrangement reaction begins with the Maillard reaction. The Maillard reaction begins with the formation of a Schiff base (glycosylamine) in a condensation reaction between the carbonyl group of a reducing sugar and an amino group of an amino

acid, peptide, or protein. The Schiff base then undergoes an *Amadori* rearrangement with a nucleophilic catalyst to form a more stable 1-amino-1-deoxy-2-ketose. This is called the *Amadori* rearrangement product (ARP). FL is one of the products of the *Amadori* rearrangement. To date, FL has been found in unroasted cocoa, paprika powder, tomato paste (DW) and red pepper (DW) [76].

Our study was designed to determine whether elderberry extract and extract-derived FL can improve Leydig cell testosterone secreting function reduced by $H_2O_2$ treatment. The results showed that the production of ROS induced by $H_2O_2$ treatment was significantly reduced by elderberry extract and extract-derived FL. In this study, the effect of elderberry extract on the expression of enzymes involved in testosterone biosynthesis and on improving the decreased testosterone secretion in $H_2O_2$-treated Leydig cells was investigated (Fig 7). The results showed that elderberry extract significantly reduced the intracellular ROS production induced by $H_2O_2$ treatment, had a positive effect on testosterone secretion, and increased the mRNA expression of genes encoding enzymes involved in testosterone biosynthesis. These results were attributed to the antioxidant and steroidogenesis-enhancing effects of the phytochemicals and bioactive substances contained in elderberry extract [77, 78]. Moreover, the results of FL treatment of $H_2O_2$-treated TM3 cells, which decreased testosterone secretion, were similar to those of elderberry extract treatment. Namely, the production of $H_2O_2$-induced ROS was significantly reduced by FL. In addition, treatment with FL increased the gene and protein expression of enzymes involved in steroidogenesis, which in turn led to increased testosterone secretion. It has been suggested that elderberry extract and FL derived from elderberry extract act as potent oxidants of $H_2O_2$-induced ROS that are generated in the oxidative stress response. This suggests that elderberry extract and FL may contribute to the protection of Leydig cells from ROS by acting as intracellular antioxidants, which in turn may protect the enzymes involved in testosterone biosynthesis in Leydig cells. This means that elderberry extract and FL may be important in maintaining the biosynthetic function of testosterone in Leydig cells by inhibiting cellular ROS production. Furthermore, a study by Chung et al. (2018) found that *Taraxacum officinale* extract ameliorated late-onset hypogonadism (LOH) by affecting steroidogenic enzymes in mouse Leydig cells [79], and a study confirming *Eurycoma longifolia*'s potential as a natural alternative to ameliorate TDS through its role as a phytoandrogen [18] can support our findings.

A number of studies have demonstrated the potential of ARP as a biomarker in the pathophysiology of inborn errors of amino-acid metabolism, diabetes, and the aging process [80–83]. However, there have been no functional studies of FL in human health. Therefore, there is a need to challenge the research on FL in the human health. In previous studies related to the uptake of FL, compared the intestinal absorption of FL and leucine by measuring the amounts of FL and leucine in the stomach and intestine of rats fed FL and leucine [84]. The results showed that the absorption of FL was very slow, and that FL remained present in the intestine 24 h after feeding. In contrast, leucine was rapidly absorbed after ingestion and was no longer found in the stomach or intestine after 5 h. These results have a very significant consequence: for a substance to be effective, it must be present or absorbed in the body at a constant dose to have a lasting effect. Even the most potent substance would have to be consumed continuously to ensure that its potency would not diminish if it has a short residence time in the body. However, FL may be able to maintain its potency with a single dose because it remains in the stomach and intestines for more than a day after ingestion. This feature is expected to be a major advantage of functional foods in terms of both efficacy and economy. Therefore, we predicted that elderberry extract and FL would enhance steroidogenesis by ameliorating $H_2O_2$-induced mitochondrial dysfunction.

This study did not address synergistic or antagonistic effects of different elder extract components on FL. However, based on the long history of elderberry use in traditional medicine,

concerns regarding adverse effects of consuming elderberry are likely to be rather low. This is because no adverse effects have been reported between elder and other food components or other medicinal plants. However, it is important to remember that medicinal plants contain many ingredients that may act in the same way as or interfere with the action of conventional medicines. Therefore, they cannot be considered completely safe. There are no clinical studies on the interaction of elderberry constituents with drugs. However, a small number of studies have cautioned against elderberry use with antidiabetic drugs, morphine, phenobarbital, diuretics, and immunologically active drugs [85].

However, much research remains to be done before elderberry extract and extract-derived FL can be used to prevent and protect against age-related muscle loss and disease. For example, various studies need to be conducted to determine whether elderberry extract and extract-derived FL have the same improvement in Leydig cell dysfunction in animal models, the part related to the regulation of hormonal homeostasis, such as the regulation of negative feedback in the brain-anterior pituitary-testis, which is important for the production of male hormones, synergistic effects with other body systems or diseases that cannot be identified in cell experiments, and the mechanism of unexpected side effects. However, it is encouraging to see that elderberry extract and extract-derived FL ameliorated Leydig cell dysfunction by inhibiting ROS production and, in particular, improved testosterone secretion by ameliorating mitochondrial dysfunction. Furthermore, we believe this discovery will contribute to the development of functional foods for improving human health, especially aging, and the development of biotechnology industries based thereon. In addition, these findings may provide a new approach to the treatment of various diseases in other body systems related to testosterone function, in addition to the development of new therapeutic and preventive agents for TDS caused by oxidative stress or aging.

## Conclusion

Taken together, the results of this study show that $H_2O_2$-induced mitochondrial dysfunction in Leydig cells resulted in decreased steroidogenesis, which was ameliorated by treatment with elderberry extract and FL (Fig 7). Importantly, elderberry extract and FL are most involved in suppressing ROS and regulating StAR during steroidogenesis because ROS are a major trigger of mitochondrial dysfunction, which in turn induces decreased steroidogenesis. Furthermore, elderberry extract and FL are important because StAR is a key regulatory point during steroidogenesis. Finally, it is predicted that elderberry extract and extract-derived FL may ameliorate the effects of decreased testosterone secretion by improving the dysfunction of the Leydig cell. Therefore, elderberry extract and the extract-derived FL may be valuable as novel, human-safe, and natural-product-derived functional foods that can help improve TDS. However, there is still a lot of work to do in order to have definitive support for these findings. Nevertheless, there are important implications of our study. First, elderberry extract and extract-derived FL have a direct effect on improving testosterone decline caused by Leydig cell dysfunction. Second, our findings provide a new avenue for approaching the treatment and prevention of a variety of diseases that are highly correlated with the functional role of testosterone in the aging process.

## Supporting information

**S1 Fig. HPLC analysis of the establishment of rutin as an indicator in the extracts of elderberries.**
(DOCX)

**S2 Fig. Analysis of FL of elderberry extract.** A: Analysis of elderberry extract by liquid chromatography/mass spectrometry. B: Analysis of elderberry extract by multiple reaction monitoring. FL, fructose–leucine.
(DOCX)

**S3 Fig. Analysis of effective indicator components (FL) of elderberry extract by multiple reaction monitoring mass spectrometry.** FL, fructose–leucine.
(DOCX)

**S1 Table. Elderberry extraction yield.**
(DOCX)

**S2 Table. Chemical analysis of the powdered elderberry extract.**
(DOCX)

**S3 Table. Nutritional Analysis of the powdered elderberry extract.**
(DOCX)

**S4 Table. Safety analysis of elderberry extract powder.**
(DOCX)

**S5 Table. Analysis of rutin content as an indicator of elderberry extract.**
(DOCX)

**S6 Table. Analysis of the content of Fructose-leucine in the extract of elderberry.**
(DOCX)

**S7 Table. Antibody information.**
(DOCX)

## Author Contributions

**Conceptualization:** Sujung Lee, Yong-Suk Kim.

**Data curation:** Sujung Lee, Jiyeon Kim.

**Formal analysis:** Sujung Lee, Jiyeon Kim, Hyunseok Kong, Yong-Suk Kim.

**Investigation:** Sujung Lee.

**Methodology:** Sujung Lee, Jiyeon Kim, Hyunseok Kong, Yong-Suk Kim.

**Project administration:** Sujung Lee, Hyunseok Kong, Yong-Suk Kim.

**Resources:** Sujung Lee, Jiyeon Kim.

**Software:** Sujung Lee, Jiyeon Kim.

**Supervision:** Hyunseok Kong, Yong-Suk Kim.

**Validation:** Sujung Lee, Jiyeon Kim.

**Visualization:** Sujung Lee.

**Writing – original draft:** Sujung Lee, Hyunseok Kong, Yong-Suk Kim.

**Writing – review & editing:** Sujung Lee, Hyunseok Kong, Yong-Suk Kim.

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
