## [Decision Letter · Decision Letter 0]

10 Jan 2024

PONE-D-23-36301Ameliorative effects of elderberry (Sambucus nigra L.) extract and extract-derived monosaccharide-amino acid on H2O2-induced decrease in testosterone-deficiency syndrome in a TM3 Leydig cellPLOS ONE

Dear Dr. Kong,

Thank you for submitting your manuscript to PLOS ONE. After careful consideration, we feel that it has merit but does not fully meet PLOS ONE’s publication criteria as it currently stands. Therefore, we invite you to submit a revised version of the manuscript that addresses the points raised during the review process.

We look forward to receiving your revised manuscript.

Kind regards,

Abeer El Wakil, PhD

Academic Editor

PLOS ONE

Journal Requirements:

"Technology development Program(S2778672) funded by the Korea Technology & Infor-mation Promotion Agency for SMEs, TIPA."

Additional Editor Comments:

This paper combines two interesting topics – male reproductive deficiency due to age related testosterone deficiency, and the advantages that come along with natural products. The study is well designed and complex. Nevertheless, there are issues raised by the reviewers that need to be addressed.

Reviewers' comments:

Reviewer's Responses to Questions

**Comments to the Author**

1. Is the manuscript technically sound, and do the data support the conclusions?

Reviewer #1: Yes

Reviewer #2: Partly

2. Has the statistical analysis been performed appropriately and rigorously? 

Reviewer #1: Yes

Reviewer #2: Yes

3. Have the authors made all data underlying the findings in their manuscript fully available?

Reviewer #1: Yes

Reviewer #2: Yes

4. Is the manuscript presented in an intelligible fashion and written in standard English?

Reviewer #1: Yes

Reviewer #2: Yes

5. Review Comments to the Author

Reviewer #1: 1. Could you mark for p-value not as different letters at every graph for showing the significancy about results? The significant codes had better show the EB or FL effect on Leydig cells than the different letters.

2. In cell viability result, you investigated the result from only 24hrs. Didn't you try to detect the cell viability with other hours? Why did you decide only 24hrs results in cell viability test?

3. Your schematic representation of the effects of EB or FL in Fig.7 is good. In Fig. 7, you mentioned about CYP19, didn't you check CYP19 protein level in this study. In supplementary data, you already showed the Cyp19 mRNA levels.

Reviewer #2: This paper combines two interesting topics – male reproductive deficiency due to age related testosterone deficiency, and the advantages that come along with natural products. The study is well designed and complex. Nevertheless, there are issues that need to be addressed:

- Since the study evaluates both the elderberry extract as well as its active component fructose-leucine, both need to be addressed in the Introduction section. Fructose-leucine is really mentioned at first in the Results. Specific information on this molecule must be provided when introducing the topic.

- Overall, the Results section only serves to describe the data gathered in the experiments. All additional information, explanations, speculations should be moved either into the Introduction or the Discussion.

- In Material and Methods, the authors describe the preparation of the extract while at the same time informing that a readily available extract was bought. Please, clarify.

- While the supplementary files contain a description of the extract, what I am missing, is a complex chemical composition of other biologically available biomolecules present in it. HPLC has obviously been performed, however only rutin is indicated. Why was rutin selected as an indicator?

- What is especially important with respect to the chemical composition is that obviously, there are other molecules in the extract that may exhibit effects that are different to fructose-leucine – meaning that there are other compounds that exhibit either beneficial or adverse properties and act in synergy or antagony with fructose-leucine. This must be properly discussed in the Discussion section.

- Overall, the Discussion section is very short and not very well written. The authors do start with explaining the chemistry of the extract and/or fructose-leucine, however the rest of the Discussion only repeats previous studies. It should contain possible explanations to the observed results, hypotheses, speculations on the mechanisms of action supported by previous studies on the male reproductive tissues and cells. Limitations and future prospects should be addressed as well.

- The Material and Methods section is lacking. The concentrations of the extract and fructose-leucine are not specified, there is no explanation on the selected concentration range. Extracts tend not to dissolve well in culture media – were the chemicals dissolved in a solvent prior to their administration to the cells? If so, was an equal amount of the solvent added to the control group? Similarly, specify what hydrogen peroxide concentration was used and why in the material and methods section.

- The cell viability and ROS assays are described very briefly – either add a relevant reference where the assays have been published or describe the assays in more detail to assure reproducibility.

- Please, add a supplementary table describing the antibodies in more detail: their isotype, clonality as well as concentration used and the blocking medium (milk or BSA).

- On a minor note, please explain abbreviations when first used in the text.

6. PLOS authors have the option to publish the peer review history of their article (what does this mean?). If published, this will include your full peer review and any attached files.

Reviewer #1: **Yes: **Hyun Joo, CHUNG

Reviewer #2: No

---

## [Author Response · Author response to Decision Letter 0]

12 Mar 2024

Dear editor

Thank you very much for reviewing my paper. 

I would like to resubmit the paper on the basis of your suggestions. I look forward to your review again. Thank you.

Dear reviewer#1

Thank you for reviewing my paper. 

I would be grateful for a review of my paper with the changes.

Dear reviewer#2

Thank you for reviewing my paper. 

Especially in the discussion section many improvements and corrections have been made. 

I would be grateful for a review of my paper with the changes.

---

## [Decision Letter · Decision Letter 1]

3 Apr 2024

Ameliorative effects of elderberry (Sambucus nigra L.) extract and extract-derived monosaccharide-amino acid on H2O2-induced decrease in testosterone-deficiency syndrome in a TM3 Leydig cell

PONE-D-23-36301R1

Dear Dr. Kong,

We’re pleased to inform you that your manuscript has been judged scientifically suitable for publication and will be formally accepted for publication once it meets all outstanding technical requirements.

Kind regards,

Abeer El Wakil, PhD

Academic Editor

PLOS ONE

Additional Editor Comments (optional):

Reviewers' comments:

Reviewer's Responses to Questions

**Comments to the Author**

1. If the authors have adequately addressed your comments raised in a previous round of review and you feel that this manuscript is now acceptable for publication, you may indicate that here to bypass the “Comments to the Author” section, enter your conflict of interest statement in the “Confidential to Editor” section, and submit your "Accept" recommendation.

Reviewer #1: All comments have been addressed

Reviewer #2: All comments have been addressed

2. Is the manuscript technically sound, and do the data support the conclusions?

Reviewer #1: Yes

Reviewer #2: Yes

3. Has the statistical analysis been performed appropriately and rigorously? 

Reviewer #1: N/A

Reviewer #2: Yes

4. Have the authors made all data underlying the findings in their manuscript fully available?

Reviewer #1: Yes

Reviewer #2: Yes

5. Is the manuscript presented in an intelligible fashion and written in standard English?

Reviewer #1: Yes

Reviewer #2: Yes

6. Review Comments to the Author

Reviewer #1: The author is answered well as my questions.

1. The significancy about result are showed wells.

2. In supplementary data, there are good results though, if your data were in line with the supplementary data,

it would be better.

I hope you have the better study about steroidogenesis with variable other extracts in the future.

Reviewer #2: The manuscript has undergone a solid revision and all comments have been incorporated into the manuscript. I have no further questions.

7. PLOS authors have the option to publish the peer review history of their article (what does this mean?). If published, this will include your full peer review and any attached files.

Reviewer #1: **Yes: **Hyun Joo CHUNG

Reviewer #2: No

---

## [Editor Report · Acceptance letter]

5 Apr 2024

PONE-D-23-36301R1 

PLOS ONE

Dear Dr. Kong, 

I'm pleased to inform you that your manuscript has been deemed suitable for publication in PLOS ONE. Congratulations! Your manuscript is now being handed over to our production team.

Kind regards, 

on behalf of

Professor Abeer El Wakil 

Academic Editor

PLOS ONE